# Chaos-Assisted Tunneling

**DOI:** 10.3390/e26020144

**Published:** 2024-02-07

**Authors:** Linda E. Reichl

**Affiliations:** Center for Complex Quantum Systems, Department of Physics, The University of Texas at Austin, Austin, TX 78712, USA; reichl@mail.utexas.edu

**Keywords:** chaos assisted tunneling, chaos, cold atoms, microlasers

## Abstract

The ability of particles to “tunnel” through potential energy barriers is a purely quantum phenomenon. A classical particle in a symmetric double-well potential, with energy below the potential barrier, will be trapped on one side of the potential well. A quantum particle, however, can sit on both sides, in either a symmetric state or an antisymmetric state. An analogous phenomenon occurs in conservative classical systems with two degrees of freedom and no potential barriers. If only the energy is conserved, the phase space will be a mixture of regular “islands” embedded in a sea of chaos. Classically, a particle sitting in one regular island cannot reach another symmetrically located regular island when the islands are separated by chaos. However, a quantum particle can sit on both regular islands, in symmetric and antisymmetric states, due to *chaos-assisted tunneling.* Here, we give an overview of the theory and recent experimental observations of this phenomenon.

## 1. Introduction

The field of classical dynamics was placed on a solid mathematical footing by the publication of Newton’s Principia in 1686 [1]. It led to the growth of science and gave support to a belief that the world is deterministic. This view was subsequently challenged by the work of Poincaré [2], who showed that perturbation expansions of dynamical processes often diverge due to nonlinear resonances (and chaos), making long-time predictions impossible.

With the development of computers in the 20th century, it became possible to explore, numerically, the consequences of chaos in dynamical systems, although scientists in the 1800’s unknowingly had begun the process with their work on thermodynamics. The so-called “ideal gas” equation of state, $PV=Nk_{B}T$ (*P* is pressure, *V* is volume, *N* is the number of particles ($\approx 10^{23}$), *T* is the temperature of the gas, and $k_{B}$ is Boltzmann’s constant), describes the macroscopic behavior of a dilute hard sphere gas in a box of volume *V*. This is a consequence of the fact that a hard sphere gas is chaotic and, therefore, ergodic (all states with the same energy are equally probable) and mixing [3]. In each collision between hard sphere particles, only quantities that are protected by the symmetries of nature survive. These include the number of particles, the total momentum, and the total kinetic energy of each pair of colliding particles. As a consequence, the macroscopic behavior of the chaotic hard sphere gas can be described in terms of its pressure (momentum conservation), particle number, and temperature (average kinetic energy). The constant $k_{B}=1.38\;\times \;10^{-23}\;{\mathrm{JK}}^{-1}$ is Boltzmann’s constant and is one of the seven fundamental constants of nature. It is the “chaos” constant because chaotic systems with many degrees of freedom are thermalized systems, and $k_{B}$ is intrinsic to such systems.

The hard sphere gas is the only many-body system that has been proven (by Sinai) to be chaotic [4]. Subsequently, Shrednicki [5] derived the Maxwell–Boltzmann distribution, which is the single-particle probability distribution for an “ideal gas” based on the fact that hard sphere gases are chaotic (ergodic and mixing). The world we live in is thermalized and is largely governed by the laws of thermodynamics. But what is the mechanism by which the world of interacting particles is thermalized, since most interactions in nature are not hard-sphere-like? That is still an open question.

“Quantum chaos” is concerned with the quantum behavior of classically chaotic systems. In the 1950s, Wigner surmised that the (quantum) Hamiltonians of classically chaotic systems have energy eigenvalue spacing distributions similar to those of a Hamiltonian matrix whose matrix elements are random numbers determined by Gaussian distributions. This “surmise” has since been verified in multiple quantum systems whose underlying classical dynamics is chaotic [6].

If we consider the dynamics of systems with few degrees of freedom, the dynamics is generally a mixture of chaotic and regular orbits. This behavior of classical dynamical systems, with few degrees of freedom, was clarified by the work of Kolmogorov [7], Arnol’d [8,9], and Moser [10] (collectively called KAM). KAM theory [6] is based on the fact that, for systems with few degrees of freedom, some regions of the phase space can be described by converging perturbation expansions (using appropriate parameters). In other regions of phase space, this is impossible because the orbits are chaotic and all perturbation expansions diverge. For systems with one degree of freedom, there is no chaos. It requires two degrees of freedom to begin to see chaotic behavior in classical dynamical systems.

For systems with two degrees of freedom (2D), the dynamics is regular (no chaos) if symmetries allow two conserved quantities (like energy and angular momentum) to exist. However, if only one symmetry governs the dynamics, it can undergo a transition to chaos as parameters are varied [11]. The quantum mechanical behavior of such systems can exhibit behavior qualitatively different from that of their classical counterpart. When the dynamics consists of a mixture of chaotic and regular orbits (KAM tori), particles that would be trapped in one region of the classical phase space can tunnel through the chaotic regions into classically prohibited regions of the phase space in the quantum system. This phenomenon is known as “chaos-assisted tunneling” or “CAT”.

Some examples of 2D systems that undergo a transition to chaos, include anharmonic potentials and hard wall asymmetric billiards. Systems with one spatial degree of freedom, driven by a time-periodic force are 1.5D systems and can also exhibit a transition to chaos. As we will describe in subsequent sections, these systems provide important platforms for observing chaos-assisted tunneling, both theoretically and in experiments.

## 2. Chaos-Assisted Tunneling in 2D Systems

Tunneling is a quantum effect that occurs, for example, in *symmetric* 1D double-well potentials. Classically, a particle placed in the left side of the double-well potential (below the barrier) will stay on that side of the potential energy barrier forever. A quantum particle, however, after it is placed in the left well, will oscillate back and forth between the wells. In a symmetric quantum double-well potential, the energy eigenstates of particles trapped below the barrier consist of symmetric and anti-symmetric energy probability amplitudes with probability equally distributed in the two wells. The energies of the symmetric and anti-symmetric eigenstates differ by a small amount, called the energy splitting $\delta =|E_{-}-E_{+}|$, where $E_{+}$ ($E_{-}$) is the energy of the symmetric (antisymmetric) state (see Figure 1a). The energy splitting, $\delta \;\approx \;e^{-A/\hbar }$ (*A* has units of action and depends on the shape of the potential) has an exponential dependence on Planck’s constant, *ℏ*. As ℏ→0 and we tend to the classical limit, the splitting disappears [12,13].

### 2.1. Anharmonic Oscillators

The concept of “dynamical tunneling” was first made explicit by Davis and Heller [12,14] (and further clarified by other authors [15,16]), when they explored the quantum behavior of a particle confined to a 2D anharmonic potential with Hamiltonian
(1)$$\begin{array}{c} H=\frac{p_{s}^{2}}{2}+\frac{p_{u}^{2}}{2}+\frac{1}{2}\omega_{s}^{2}s^{2}+\frac{1}{2}\omega_{u}^{2}u^{2}+\lambda u^{2}s+\beta s^{2}u \end{array}$$ The potential in Equation (Equation 1) has no potential energy barriers, but energy contours have a triangle-like shape which, for β=0, is symmetric about u=0 (see sketch in the upper part of Figure 1b). For β=0, the classical system has pairs of periodic orbits that have the same energy and are mirror images of each other (after reflection through u=0). They bounce back and forth between opposite walls (in the u direction) of the potential (orbit “A” and “B” in Figure 1b). Classically, they can be identified with stable fixed points in a $p_{u}$ versus *u* surface of section of the classical phase space (see the lower part of Figure 1b).

The quantum version of the symmetric (β=0) anharmonic oscillator system has energy eigenstates that form symmetric and antisymmetric pairs, with slightly different energies, whose probability distributions sit along both classical periodic orbits (A and B in Figure 1b). The absolute value of one of the symmetric eigenstates is shown in Figure 2a. This behavior of eigenstates in the quartic potential is similar to the symmetric and antisymmetric pairs of eigenstates in the symmetric double-well system, except for the fact in the quartic potential there is no potential energy barrier to tunnel through. When β≠0 and the reflection symmetry is broken, and the eigenstates again lie primarily along only one of the pair of classical periodic orbits in Figure 1b. Davis and Heller [12] proposed *chaos-assisted tunneling* as a mechanism to allow the system to form symmetric and antisymmetric pairs of energy eigenstates, even though there is no potential barrier to tunnel through. From Figure 2c, we see that there is a vast region of chaos separating the stable islands that support the energy eigenstate in Figure 2a.

In order to confirm the concept of chaos-assisted tunneling, Tomsovic and Ullmo [13] *modeled* the quartic system in terms of symmetric $\Psi_{R}^{+}$ and antisymmetric $\Psi_{R}^{-}$ regular states, each with energy $E_{R}$ (the energies they would have without the effect of tunneling), that were isolated on the stable islands at $p=\pm p_{o}$. For one case they considered, they assumed the region of chaos separating the two stable islands had no blockages (for example no cantori partially blocking classical trajectories [6]). They assumed that the states in the chaotic region were either symmetric or antisymmetric. Then, for each symmetry class, they *modeled* the dynamics in terms of a Hamiltonian that coupled the state in the regular region to the states in the chaotic region. As the energies of the states were varied with variation in a parameter (such as λ in Equation (Equation 1)), they found that the coupling *v* between the regular and chaotic states was enhanced when the energies associated with the regular and chaotic states undergo avoided crossings.

Since it was not possible to actually construct the states in the chaotic region accurately, they assumed that the chaotic states had energy eigenvalues govern by the Gaussian orthogonal ensemble (GOE) and that the coupling $v_{n}$ between the nth chaotic state and the regular state was a Gaussian random variable with variance $v^{2}$. For the case where the chaotic sea has no partial blockages, the Hamiltonian they constructed (for the symmetric states) had a structure
(2)$$\begin{array}{c} H^{+}=\left(\begin{array}{cc} E_{R} & {\bar{v}}_{+}\\ {\bar{v}}_{+}^{T} & {\bar{\bar{E}}}_{GOE}^{+} \end{array}\right), \end{array}$$
where ${\bar{\bar{E}}}_{GOE}^{+}$ is a matrix of energies of the chaotic states (given by the Gaussian orthogonal ensemble) and $\bar{v}$ is a row matrix of Gaussian random variables $v_{n}$. A similar matrix was written for the antisymmetric states. They then computed the energy splittings of the pairs of regular symmetric and antisymmetric states. The splitting distribution is given in Figure 3. Tomsovic and Ullmo also constructed random Hamiltonians for cases where there were blockages in the chaotic regions and found level splitting distributions that were close to, but not exactly the same as the case with no blockages.

Leyvraz and Ullmo [16] further investigated the statistical properties of symmetric and antisymmetric energy eigenstates (for the quartic oscillator) whose structure is the result of chaos-assisted tunneling. They found that the probability distribution of energy level splittings $\delta =\;|E_{+}-E_{-}|$ was given by a Cauchy distribution
(3)$$\begin{array}{c} P\left(\delta s\right)=\frac{2}{\pi }\frac{1}{1+\delta s^{2}}, \end{array}$$
where $\delta s=\delta /\delta_{typ}$, and $\delta_{typ}$ is a typical value of the splitting [17]. The distribution of level splittings, shown in Figure 3, when plotted in terms of appropriate variables, is consistent with the result of Leyvrz and Ullmo.

The work of all these authors, including Davis and Heller [12]; Bohigas, Tomsovic, and Ullmo, [15]; Tomsovic and Ullimo [13]; and Leyvraz and Ullmo [16], showed that energy eigenstates (where chaos-assisted tunneling plays a role due to the presence of chaos) appear to have (i) great enhancement of their average (energy) splitting; (ii) extreme sensitivity to the variation in an external parameter; and (iii) strong dependence on the tunneling properties to any blockages in the chaotic region separating the two tunneling tori.

### 2.2. Time-Periodic 1D Systems

Anharmonic systems with one spatial degree of freedom, driven by time-periodic forces, have provided another testing ground for issues related to chaos-assisted tunneling. Lin and Ballentine [18] studied the dynamics of a particle in double-well potential driven by a time periodic force,
(4)$$\begin{array}{c} H=\frac{p^{2}}{2m}+Bx^{4}-Dx^{2}+\lambda x\cos\left(\omega t\right). \end{array}$$ By analyzing the Floquet states, they found erratic tunneling times for states that are initially localized on field induced periodic orbits. They explain this in terms of overlap of the “regular” states with a large number of eigenstates states that “randomly” cover the chaotic sea. The role played by avoided crossings on the tunneling process for a periodically driven pendulum was also studied by Latka, Grigolini, and West [19].

Mechanisms for chaos-assisted tunneling in other time-periodically driven systems have been considered by several authors. For example, Roncaglia et al. [20] considered the time-periodically kicked Harper model,
(5)$$\begin{array}{c} H=K\cos\left(p\right)+K\cos\left(x\right)\sum_{n=-\infty }^{+\infty }\delta \left(t-n\right). \end{array}$$ They found that when the chaotic region and Planck’s constant were the same size and tunneling rates were irregular, which they interpreted to be a signature of the chaos present in the system.

### 2.3. Billiards

Chaos-assisted tunneling has also been explored in several different types of billiards. As mentioned earlier, symmetric billiards like the circle or half-circle billiard support integrable dynamics, but if the symmetry is broken by distorting the shape of the billiard, the dynamics becomes non-integrable and can support a variety of dynamical behaviors ranging from small regions of chaos to fully chaotic dynamics [11].

Frischat and Doron [21,22] performed extensive studies of mechanisms for chaos-assisted tunneling between states in the annular billiard, which consists of a large circle billiard, with a smaller circle cut out of it off-center (see Figure 4a). Classically, this billiard has two whispering gallery modes, which correspond to waves (or particles) that travel close to and along the outer wall due to internal reflection. One whispering gallery mode travels clockwise and the other travels counterclockwise. If the small circular cutout is placed off-center, the billiard dynamics will consist of a mixture of regular and chaotic orbits (see Figure 4b). Classically, the whispering gallery modes will not be blocked if the small circular cutout is not in contact with the walls. For this system, the whispering gallery modes play the role of the states in the double-well system. If the dynamics of the objects in the billiard are governed by wave motion (such as quantum particles or photons), then the whispering gallery modes can form symmetric or antisymmetric standing waves due to chaos-assisted tunneling involving the chaotic states in the billiard.

Hackenbroich and Nöckel [23], numerically studied the effect of chaos on the lifetimes of whispering gallery (WG) modes in an annular dielectric billiard with a metallic inclusion and found that the lifetimes of the whispering gallery modes fluctuated by orders of magnitude as the location of the small circular inclusion was varied due to avoided crossings with chaotic states in the annular billiard. Since the waves inside the dielectric billiard can decay, the system can be thought to behave like an open system with long-lived quasibound states whose dynamics are strongly affected by the regular and chaotic dynamics inside the billiard.

#### 2.3.1. Microwave Cavities

In 2000, Dembowski et al. [24] provided the first experimental confirmation of chaos-assisted tunneling in a microwave annular billiard, and it was followed by a more extensive analysis in 2005 by Hofferbert et al. [25]. They studied microwave dynamics in both a superconducting niobium resonator and in a normal conducting copper resonator. In a symmetric annular billiard (circular inset in the middle), the whispering gallery modes (clockwise and counterclockwise) have opposite angular momentum and are degenerate. However, when the symmetry is destroyed by moving the circular inset off-center, the whispering gallery modes become coupled and form symmetric and antisymmetric pairs. The energy splitting of these pairs showed the signatures of chaos-assisted tunneling in the de-symmetrized annular microwave billiard.

Several years later, Bäcker et al. [26], studied chaos-assisted tunneling using microwave spectra in a mushroom billiard. The mushroom billiard consists of a quarter circle with a rectangular billiard attached to one side so the radius and the left side of the rectangle are aligned. In this case, there are whispering gallery modes along the circular edge of the quarter circle. They tunnel between each other via avoided crossings with the chaotic states in the billiard (the chaotic states sample the whole billiard.)

#### 2.3.2. Optical Cavities

Nöckel and Stone et al. [27,28,29] studied the lifetime of whispering gallery modes for a deformed cylindrical dielectric optical resonator. They found that the deformation of the optical resonator led to highly anisotropic emission of the whispering gallery modes and significantly shortened lifetimes of the whispering gallery modes, due to avoided crossings with the chaotic states inside the resonator.

One of the most wide-spread uses of chaos-assisted tunneling (CAT) is in the field of microlasers [30]. Circular optical cavities (2d billiards containing light waves) have whispering gallery modes that emit light isotopically. In a series of papers in 1994–1997, Nöckel, Stone, and co-authors [23,28,31], analyzed the effect on light emission if the optical cavities are slightly deformed so that chaotic dynamics starts to play a roll. They found that the light emission could become highly anisotropic, so that chaos could, in principle, be used to control emission of light from the microcavities.

Since these early theory papers, a number of experiments have shown that chaos-assisted tunneling in microlasers can be used to control the performance and output of the microlasers. In 2010, Shinohara et al. [32,33], studied the performance of a deformed GaAs/AlGaAs disk cavity. By comparing experimental data to the results of numerical simulations that show significant chaos in the phase space of the deformed cavity, they were able to show signatures of chaos in the light emission pattern of the microlaser. In 2013, several groups observed chaos-assisted tunneling in microlasers. Kim et al. [34] demonstrated chaos-assisted tunneling in a rounded half-moon shaped InGaAsP semiconductor microcavity laser. Xiao et al. [35] showed that chaos-assisted tunneling in the phase space of a slightly deformed optical microcavity can give rise to a new form of induced transparency in the device.

In 2021, Qian et al. [36], considered a microcavity with a quadrapole shape whose radius could be written $R\left(\phi \right)=R_{0}\left[1+\epsilon \cos\left(2\phi \right)\right]$, where $R_{0}$ = 10 μm is the approximate radius and ϵ=0.12. In 2021, Wang et al. [37] showed that it is possible to map the light wave mode patterns in an optical microcavity and showed the existence of chaos-assisted tunneling in a silicon microdisk “with unprecedented certainty”.

### 2.4. Cold Atoms

Since the dynamics of atoms is governed by the Schrodinger equation, which is a wave equation, the atomic dynamics is described in terms of complex “probability” waves. Atom-optics experiments make use of the wave nature of atoms and, so far, have used alkali atoms, either sodium (Na) or cesium (Ce). The alkali atoms are made to interact with a standing wave of laser light (formed from two counter-propagating laser beams). The two counter-propagating laser beams form a periodic standing wave. The standing wave of light is slightly detuned away from resonance with a specific pair of atomic energy levels with energy spacing $\hbar \omega_{0}$. The light wave stimulates absorption and emission of a photon by the atoms, which results in a net atomic recoil of $2\hbar k_{L}$ ($k_{L}=\frac{\omega_{L}}{c}$ is the wave vector of the laser beam). When the laser detuning, $\delta_{L}=\omega_{0}-\omega_{L}$, is large, this process dominates the dynamics.

The theoretical model that describes the atom-optic experiment was originally developed by Graham, Schlautman, and Zoller [38]. In 2001, two different experimental groups, one in Texas (Steck, Oskay, and Raizen [39,40,41] and the other at NIST (Hensinger et al. [41,42,43], observed chaos-assisted tunneling in atom-optics experiments (here, we follow the theoretical analysis in [44,45]).

The Texas experiment involved (after considerable manipulation [39]) a cloud of about $10^{4}$ cesium atoms, with fairly well-defined momentum at a temperature of about $T=4\;\times \;10^{-7}$ K. The Hamiltonian that described the center-of-mass dynamics of each cesium atom in the oscillating standing wave of light is
(6)$$\begin{array}{c} H_{1}=\frac{p_{1}^{2}}{2m}-2V_{o}\cos^{2}\left(\frac{\omega_{m}t}{2}\right)\cos\left(2k_{L}x_{1}\right), \end{array}$$
where $\omega_{m}=\frac{2\pi }{T}$ is the modulation frequency of the light (and T=20 μs the period), $V_{o}=\frac{E_{o}^{2}d^{2}}{2\hbar \delta_{L}}$, $E_{o}$ is the electric field amplitude, and *d* is the dipole moment of the cesium atom. Interaction with the laser beam causes absorption, and then emission, of a photon causing a cesium atom a momentum change of $2\hbar k_{L}$. Thus, the momentum of the cesium atoms is quantized in units of $2\hbar k_{L}$ and we can use Floquet (not Floquet–Bloch) theory to describe the dynamics (Floquet–Bloch theory was used in [46,47]). In dimensionless units, the Hamiltonian can be written
(7)$$\begin{array}{c} H=\frac{p^{2}}{2}-2\alpha \cos^{2}\left(\pi \tau \right)\cos\left(\phi \right), \end{array}$$
where $\phi =2k_{L}x_{1}$, $\tau =\frac{t}{T}$, $p=\frac{4\pi k_{L}p_{1}}{m\omega_{m}}$, and $H=\frac{16\pi^{2}k_{L}^{2}}{m\omega_{m}^{2}}$. Strobe plots of the classical phase space are shown in Figure 5 for α=2 and α=10. Theoretical curves for momentum oscillations, as a function of time, are given in Figure 6 for α=2.0 and α=10. The experimental curve for momentum oscillations for α=9.7 is given in Figure 7. Note that α depends inversely on Planck’s constant. For α=2.0, there is no tunneling between the states localized on the large islands at p≈±3 in Figure 5. For α=9.7, tunneling has occurred and the quantum system can now “see” the chaotic region separating the two islands at p≈±3. As pointed out in [45,47], for the parameters used in the experiment, the effective Planck’s constant is too large for the experiment to see the tunneling complexity that occurs when a number of states lie in the chaotic sea. However, the experiment is seeing chaos-assisted tunneling, but not all the complexity that can come with it.

The Hamiltonian describing the Hensinger experiment [42],
(8)$$\begin{array}{c} H=\frac{p^{2}}{2}+2\kappa \left[1+2\epsilon \sin\left(\omega t\right)\right]\sin^{2}\left(x/2\right), \end{array}$$
differs from that of the Raizen experiment by the placement of resonant islands in the chaotic sea, but the analysis is similar.

The experiments of Raizen et al. and of Hensinger et al. were not far enough into the semiclassical regime to see all the complexity associated with signatures of chaos-assisted tunneling. In 2016, Dubertrand et al. [17] did a more extensive analysis of typical cold atom experiments and, through numerical simulations, obtained the important signatures of chaos-assisted tunneling. They considered the dimensionless Hamiltonian
(9)$$\begin{array}{c} H=\frac{p^{2}}{2m}-\frac{U_{0}}{2}\left(1+\epsilon \cos\left(\omega t\right)\right)\cos\left(\frac{2\pi x}{d}\right), \end{array}$$
where *m* is the mass of the alkali atoms used in the experiment, ω is the modulation frequency of the light wave, ϵ is the amplitude of the time-periodic modulation, $U_{0}$ is the depth of the periodic standing wave of light for ϵ=0, and *d* is the spatial period of the optical lattice. If we now introduce dimensionless variables
(10)$$\begin{array}{c} p=\frac{2\pi }{m\omega d}p,\;\;\;x=\frac{2\pi }{d}x,\;\;\;t=\omega t, \end{array}$$
the dimensionless Hamiltonian takes the form
(11)$$\begin{array}{c} H=\frac{p^{2}}{2}-\gamma \left(1+\epsilon \cos\left(t\right)\right)\cos\left(x\right), \end{array}$$
where
(12)$$\begin{array}{c} E_{L}=\frac{h^{2}}{2md^{2}},\;\;\;\gamma ={\left(\frac{E_{L}}{\hbar \omega }\right)}^{2}s,\;\;\;s=\frac{U_{0}}{E_{L}},\;\;\;\mathrm{and}\;\;\;\hbar_{eff}=\frac{2E_{L}}{\hbar \omega }. \end{array}$$ Here, $E_{L}$ is a characteristic energy scale for the optical lattice, *s* is a dimensionless depth of the lattice potential, and $\hbar_{eff}$ is an effective Planck’s constant for the dimensionless system. They studied the classical dynamics of this model system. They computed the Lyapunov exponents of classical trajectories and found parameter regimes with significant chaos but also large regular islands at $p=\pm p_{o}$. Dubertrand et al. [17] could identify the symmetric and antisymmetric pairs of states that sit in the regular islands at $p=\pm p_{o}$ and computed how their energy splitting $\delta =|E_{+}-E_{-}|$ varied. They define a fractional splitting $\delta s=\frac{\delta }{\delta_{typ}}$, where $\delta_{typ}$ is a “typical” splitting. They obtain the distribution of splittings as $\hbar_{eff}$ is varied and compared it to the Cauchy distribution obtained by Leyvraz and Ullmo [16] based on GOE. A plot of the actual splittings is shown in Figure 8 and a comparison of the actual splitting distribution and the theoretical Cauchy distribution is shown in Figure 9. The data obtained from the cold atom Hamiltonian gives good agreement with GOE predictions.

One of the conclusions of the analysis of Dubertrand et al. [17] is that it might not be feasible, given the experimental constraints of the atom-optics experiments, to see all the complexity of chaos-assisted tunneling between symmetric island pairs separated in *momentum* space. They proposed a series of steps and changes in parameters that would allow the central island, in Figure 5a, to undergo bifurcation into two symmetric islands separated in *position* space. They then indicate that the scheme they propose might be feasible with current technology.

In 2020, Arnal et al. [48] again pointed out that atom cooling techniques are extremely versatile and allow the modeling of a variety of systems where the wave nature of atomic motion is essential. A variety of potentials can be produced to influence atomic motion. They can mimic situations commonly found in condensed matter systems. They then analyze the dynamics of atom optic system with a Hamiltonian very similar to that used in the Raizen experiment.

## 3. Three or More Degrees of Freedom

It is not clear if dynamical tunneling can play the same role in nD systems (n≥3) in regard to the control of wave dynamics. For example, for systems with 3D, the “landscape” totally changes. In 1963, Arnold showed that conservative classical systems with three or more degrees of freedom are intrinsically unstable [8,9]. For such systems, the energy surface is covered densely by interconnected resonance lines (an Arnold web) and the system can diffuse throughout the “energy surface” in the high-dimensional phase space [6,49,50].

To understand the complexity of the classical dynamics in 3D, consider the anharmonic, time-periodic lattice (effectively a 3D system)
(13)$$\begin{array}{c} H\left(x,y,t\right)=p_{x}^{2}+p_{y}^{2}+\left(V_{0}+V_{1}\cos^{2}\left(\omega t\right)\right]V\left(x,y\right), \end{array}$$
where
(14)$$\begin{array}{c} V\left(x,y\right)=U[\cos^{2}\left(x\right)+\cos^{2}\left(y\right)+b\cos\left(x\right)\cos\left(y\right)]. \end{array}$$ For b=0, the dynamics in the x- and y-directions is uncoupled. The dynamics in the x-direction behaves as if it was governed by the Hamiltonian
(15)$$\begin{array}{c} H\left(x,t\right)=p_{x}^{2}+p_{y}^{2}+\left(V_{0}+V_{1}U\cos^{2}\left(\omega t\right)\right]\cos^{2}\left(x\right). \end{array}$$ In Figure 10a, we show a strobe plot of the dynamics for b=0. We see a region of chaos surround by KAM tori. However, when b≠0, the dynamics fundamentally changes. A strobe plot for b≠0 is shown in Figure 10b. As discussed in [49], the dynamics is governed by a dense Arnold web and diffusion of the system trajectories now occurs throughout the 3D phase space.

It would be very interesting to look for the phenomenon of chaos-assisted tunneling in these higher dimensional systems (3D or more). It would likely have to occur in parameter regimes where most of the phase space still consists of non-resonant KAM tori [6,49]. However, at this time, the quantum dynamics of such systems still remains largely unexplored.

## 4. Conclusions

For 2D systems with two conserved quantities, such as energy and angular momentum for a circular billiard, the classical and quantum dynamics is regular (integrable). However, if one of the symmetries is broken (deform the circle), the dynamics becomes a mixture of chaotic and regular orbits. A classical particle in a regular region is trapped there forever, but a quantum particle (like an atom or a photon) can tunnel through the chaotic region. The first theoretical discussion of this phenomenon is due to Davis and Heller [12] in 1981. The work of Davis and Heller inspired numerous subsequent theoretical contributions to the field by other authors (many described in sections above and also in the review [41]).

The first experimental observation of chaos-assisted tunneling occurred in microwave cavities in 2000 [24]. That was followed, in 2001, by observation of chaos-assisted tunneling in cold atomic systems (alkali atoms trapped in standing waves of light) [39,40,42,43]. In 2010, signatures of chaos-assisted tunneling were found in microlasers [32,33]. Subsequent to those early experiments, it has proven to be an important and practical tool for controlling radiation emitted from microlasers of various shapes.

It has been suggested that it can play a role in the internal dynamics of molecules [14,51]. It has been observed recently in the dynamics of periodically kicked spin systems [52]. Indeed, chaos-assisted tunneling has been shown to play a key role in the quantum dynamics of many different physical systems with two degrees of freedom, when they can be shown to have a classical counterpart. It will be extremely interesting to see if situations occur where it governs aspects of the quantum dynamics of systems with three or more degrees of freedom.

## Figures and Tables

**Figure 1 entropy-26-00144-f001:**
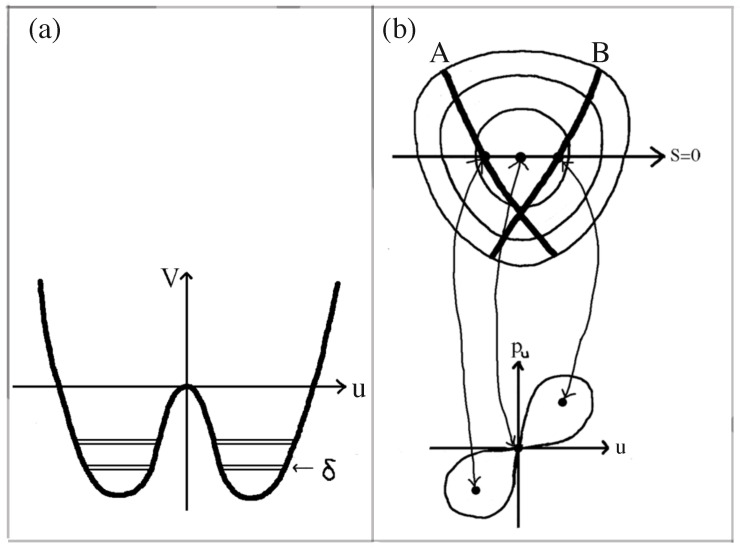
(**a**) A double-well potential with a potential energy barrier. (**b**) Periodic orbits (A and B) of the quartic potential with no potential energy barriers (Based on [12]).

**Figure 2 entropy-26-00144-f002:**
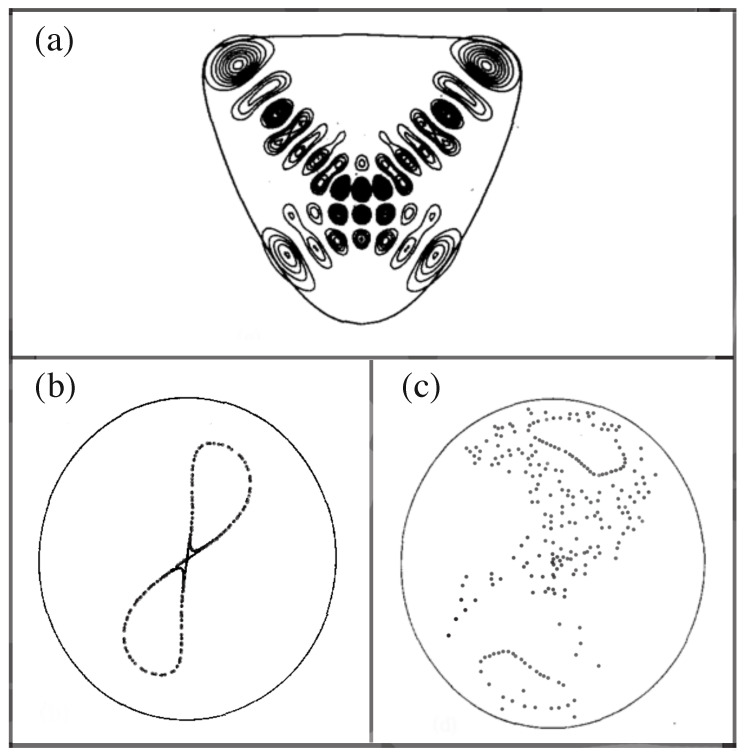
Some aspects of the quantum and classical dynamics of the quartic oscillator Equation (Equation 1) for parameters $\omega_{s}=1.0$, $\omega_{u}=1.1$, $\lambda_{s}=-0.11$, and β=0. (**a**) The symmetric eigenstate with energy E=13.59. (**b**) Surface of section, $p_{u}$ versus *u*, for energy E=9.0. (**c**) Surface of section, $p_{u}$ versus *u*, for energy E=13.6. The circular curves in (**b**,**c**) are energy boundaries. (Reproduced from [12] with permission of AIP publishing).

**Figure 3 entropy-26-00144-f003:**
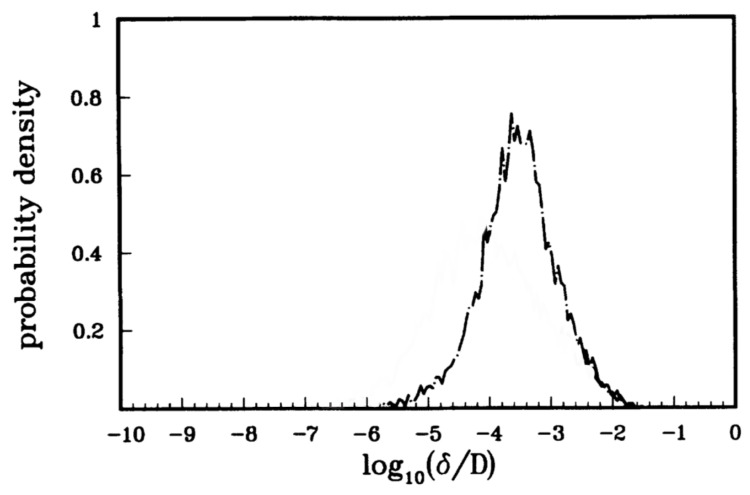
Level splittings for quartic potential. (Reprinted Figure 10 from [13] with permission from the American Physical Society).

**Figure 4 entropy-26-00144-f004:**
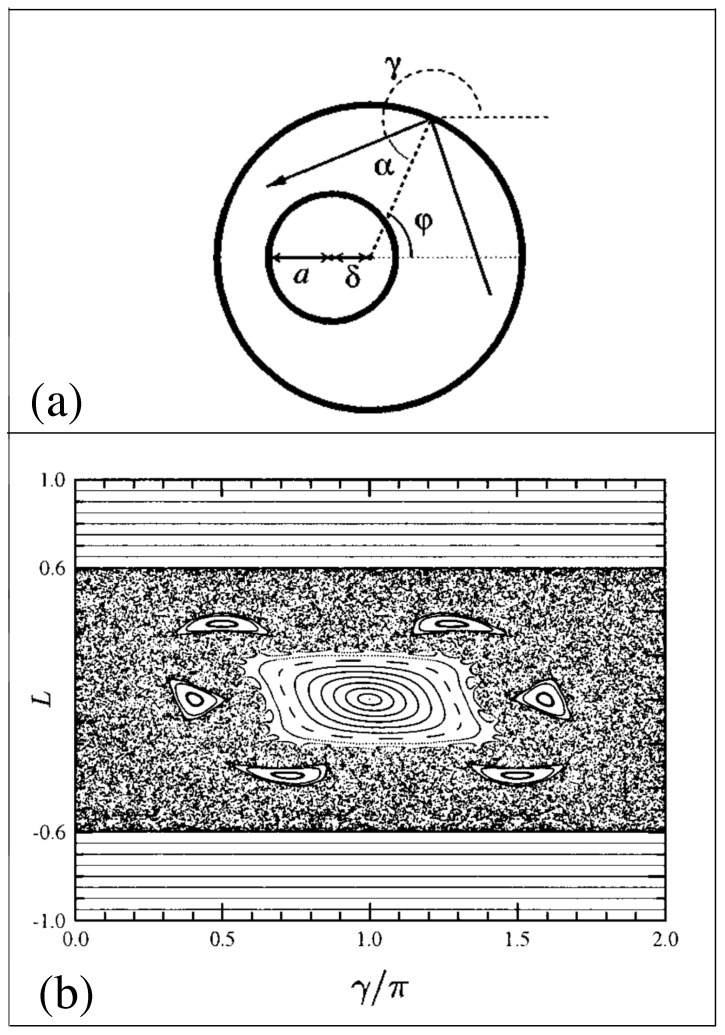
(**a**) Annular billiard. (**b**) Surface of section (angular momentum *L* versus γ/π) of orbits in the annular billiard for a=0.4 and δ=0.2 (the angle γ is shown in (4a)). (Reprinted Figures 1 and 2 from [22] with permission from the American Physical Society).

**Figure 5 entropy-26-00144-f005:**
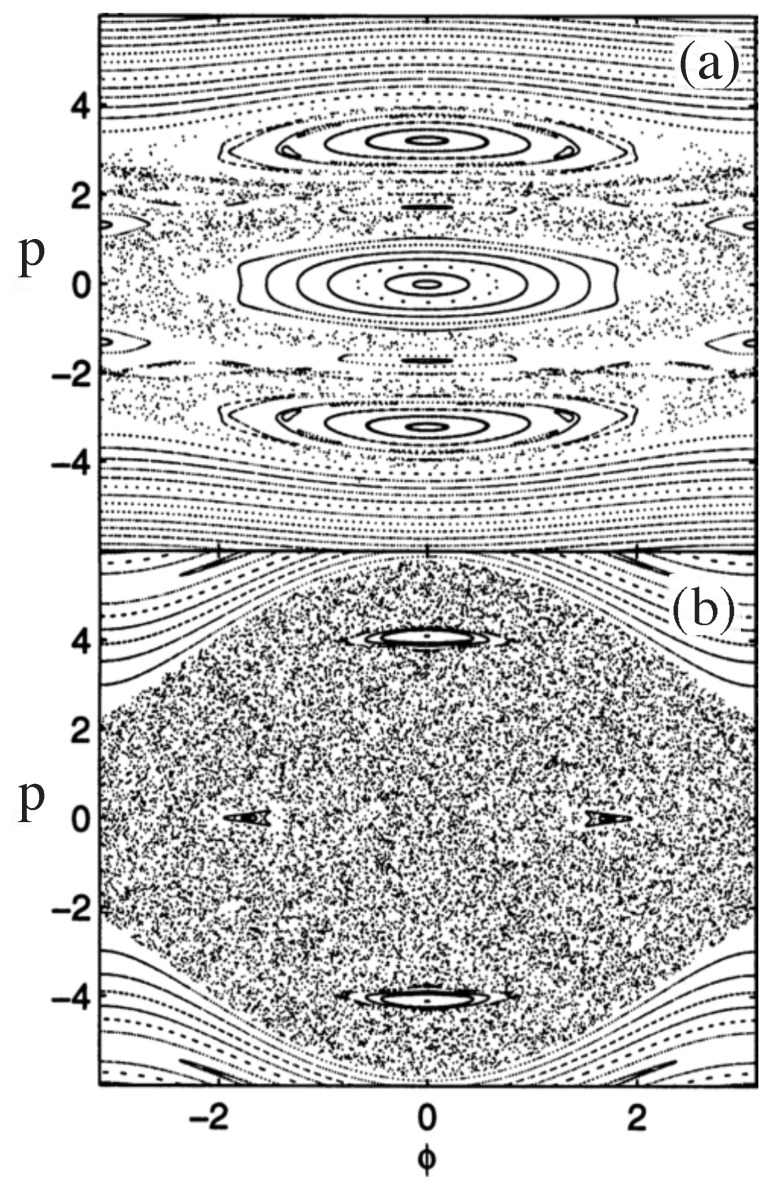
Strobe plots of cold atoms governed by Equation (Equation 7). (**a**) α=2. (**b**) α=10. (Reprinted Figure 1 from [45] with permission from World Scientific Pub.).

**Figure 6 entropy-26-00144-f006:**
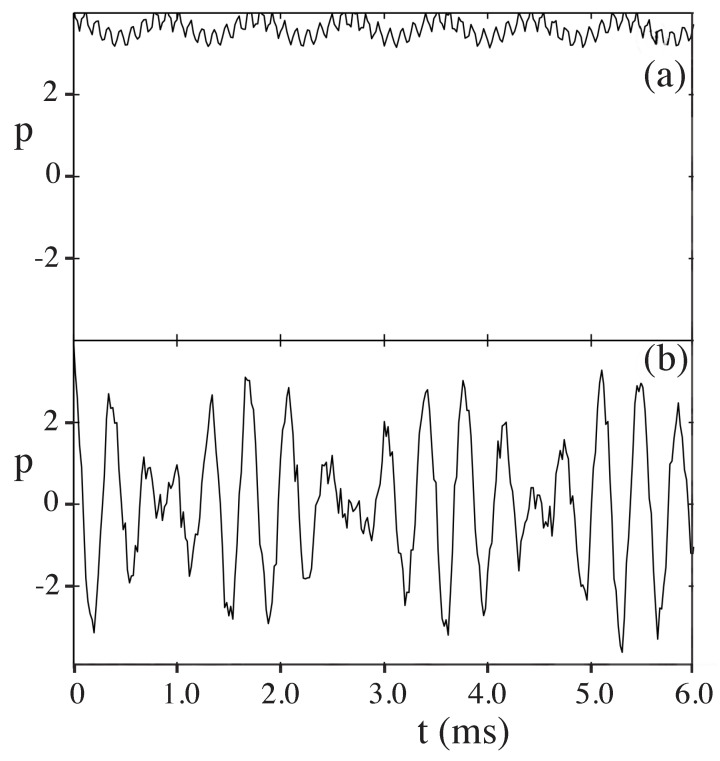
Time evolution of the momentum expectation value for cold atoms governed by Equation (Equation 7). (**a**) For α=2, no chaos-assisted tunneling. (**b**) For α=10, chaos-assisted tunneling occurs. (Reprinted Figure 2 from [45] with permission from World Scientific Pub.)

**Figure 7 entropy-26-00144-f007:**
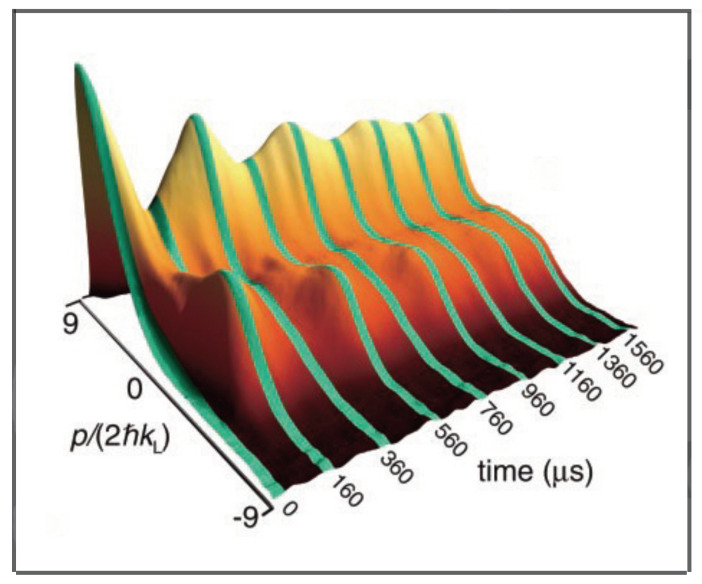
Experimentally measured oscillation of momentum distribution of cold atoms (governed by a Hamiltonian-like Equation (Equation 7)), as a function of time. The initial distribution was centered in the upper regular island. The momentum oscillates between the upper and lower regular islands. (Reprinted Figure 1 from [39] with permission from the American Association for the Advancement of Science.)

**Figure 8 entropy-26-00144-f008:**
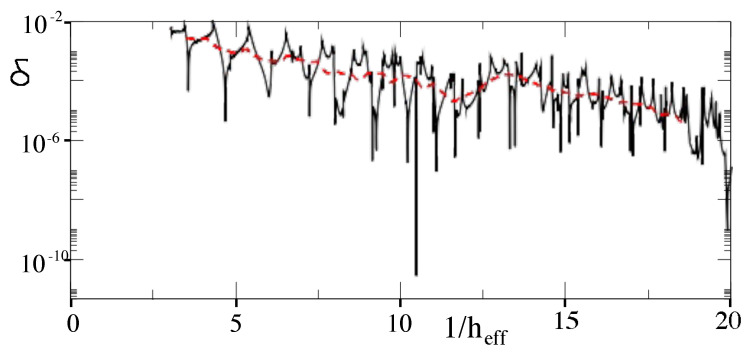
Energy splitting, δ, of symmetric and antisymmetric states as a function of $1/\hbar_{eff}$ for ϵ=0.4 and γ=0.25. The red dashed line is the “typical value”. (Reprinted Figure 4a from [17] with permission from the American Physical Society.)

**Figure 9 entropy-26-00144-f009:**
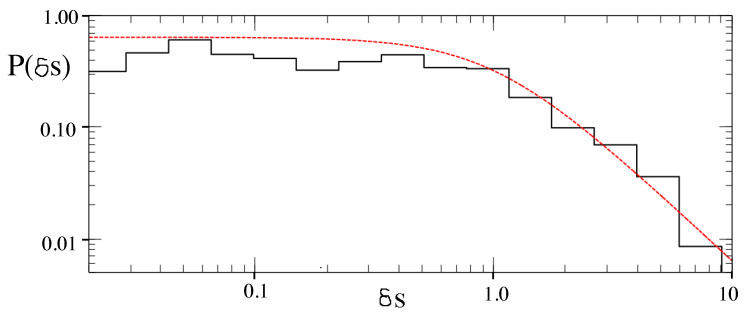
Probability distribution of energy splittings, δs, for the Hamiltonian in Equation (Equation 11) (see Figure 8). The red dashed line is the prediction of Equation (Equation 3). (Reprinted Figure 4b from [17] with permission from the American Physical Society.)

**Figure 10 entropy-26-00144-f010:**
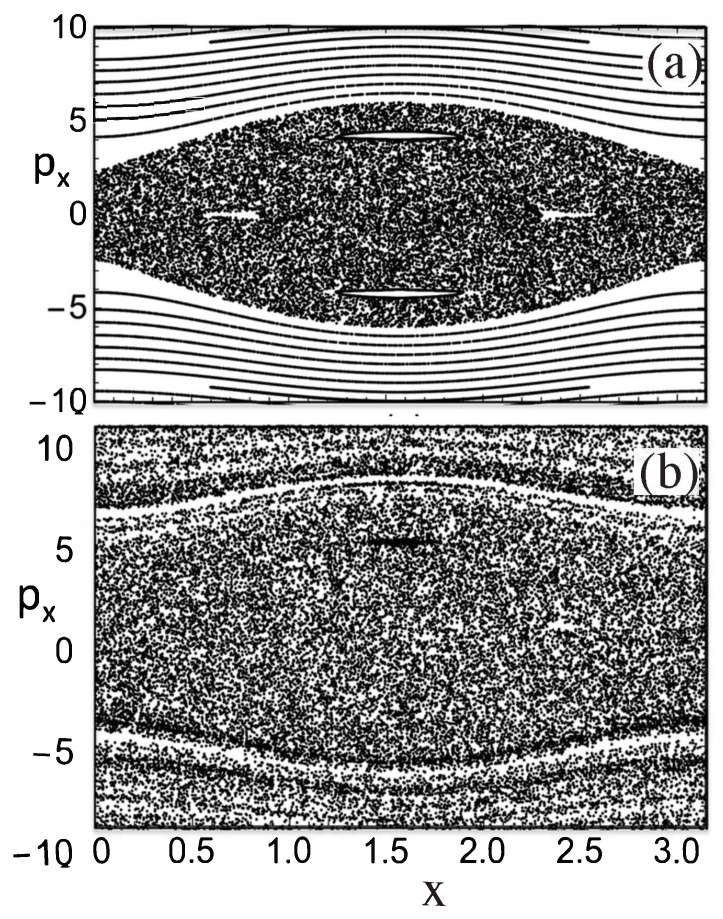
Dynamics of governed by Hamiltonian in Equations (Equation 13) and (Equation 14). Strobe plot of $p_{x}$ versus *x* for each period of the field. (**a**) b=0. (**b**) b=0.002. (Reprinted Figure 3 from [49] with permission from the American Physical Society).

## Data Availability

No new data were created or analyzed in this study. Data sharing is not applicable to this article.

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
