# Peer review of "Chaos-Assisted Tunneling"

_entropy, 2024, doi:10.3390/e26020144_

Round 1
Reviewer 1 Report
Comments and Suggestions for Authors
Chaos-assisted tunneling is a particularly appealing phenomenon within
the field of quantum chaos, as it combines peculiarities of complex
classical dynamics with an emblematic quantum effect, tunneling, and
is manifest in several of the known quantum signatures of classical
chaos, such as spectral statistics and phase-space representations of
quantum eigenstates. After more than two decades since it has been
pointed out for the first time and is now rather comprehensively
explored by theoretical as well as experimental studies, time is ripe
for a review, as provided by this paper.
The author offers a brief yet well-balanced overview over the most
relevant developments in theory and experiment in the subject,
illustrated by a number of pertinent examples, complemented by
demonstrative figures and a selection of the most important
references. This review certainly represents a valuable contribution
to Entropy.
Yet there is space for improvement. In the sequel, I list several
cases where I think details should be amended or a modification could
increase the impact of the paper, from conceptual issues down to
technical details, following the order of their appearance in the
text:
– Introducing a review with a broad historical retrospect is certainly
welcome. Yet I am concerned about a specific point made in Section 1.
It features the importance of chaos for the foundations of statistical
mechanics, focusing on ergodicity. While it is decisive for the
interchangeability of ensemble means and time averages, the relevant
condition for thermalization and equilibrium, is the property of
mixing. Unlike ergodicity, mixing is sufficient for the approach to
equilibrium and requires chaotic dynamics in turn. Mixing marks the
next higher level of dynamical irregularity above ergodicity, see,
e.g., my Ref. [1] below. Indeed, the breakthrough achieved by the work
of Ya. G. Sinai was the proof that the 2-dim. billiard bearing his
name, and with it the hard-sphere gas mentioned in the paper, is
mixing. The respective roles of ergodicity and mixing should be kept
clear in this introduction.
– Section 1 ends with a short outline of chaotic dynamics in classical
Hamiltonian systems. However, it does not address the subject proper
of this paper, chaos-assisted tunneling. While it is of course
elucidated in detail in the subsequent sections, readers could expect
al least a brief qualitative summary what chaos-assisted tunneling is
about.
– The first line after Eq. (1) says that "The potential in Eq. (1) has
no potential energy barriers, but has a triangle-like shape". That is
misleading. It is the contour lines of the Hamiltonian, shown in Fig.
1, that appear triangular, while the potential along any cross
section of the two-dimensional position space is a quadratic
function.
– Figure 1 shows a schematic double-well potential as panel (a) and
the phase space corresponding to Eq. (1) with two periodic orbits as
panel (b) of the same figure. This immediate neighbourhood could
insinuate that the phase space in (b) pertains to the potential
displayed in (a), while in fact it is based in a potential without
barrier. That should be made clear in the figure caption.
– Triple avoided crossings are at the heart of chaos-assisted
tunneling, which include the energies a pair of eigenstates of
different dynamic nature (one chaotic, one regular) with the
same parity, therefore forming an avoided crossing where they
intersect, and that of a third state of the same structure as the
regular member of the avoided crossing, but opposite parity. The way
this somewhat intricate configuration is described in the text is
correct, but not very graphical. An illustrating figure would be
extremely helpful, either as a schematic drawing or based on real
data. An instructive pair of plots can be found in Ref. [2] below,
Figs. (1b) and (4) (I suggest to include this paper in the
bibliography anyway). Such a figure could also explain what otherwise
remains somewhat unmotivated, namely the statement that "the coupling
v between the regular and chaotic states is enhanced when the energies
associated with the regular and chaotic states undergo avoided
crossings".
– In the context of an approach to chaos-assisted tunneling from
spectral statistics, the paper talks of a "quartic system in terms of
symmetric \Psi_R^+ and antisymmetric \Psi_R^- regular states, each
with energy E_R". If this is a tunneling doublet, shouldn't there be
a non-zero energy gap, so that E_R^- > E_R^+ ?
– In Fig. 2, panels (b) and (c) are specified as "surfaces of
section", without saying, sections of what. I would interpret these
plots as stroboscopic traces of trajectories (panel (c) presumably
does not show a periodic orbit). Complementary question: Whence the
circular outlines of these plots?
– Quite evidently, the distribution of level splittings shown in
Fig. 3 serves to illustrate the theoretical expression, Eq. (3).
However, this figure is not referenced in that context.
– In the context of billiards, Section 2.3, the paper addresses
lifetimes of whispering-gallery modes. That would indicate that these
modes amount to metastable states. How can decay arise in a closed
Hamiltonian system?
– It appears that Figs. 5, 6, and 7 refer to the cold-atom system
modeled by the Hamiltonian given in Eq. (5). Please mention this in
the captions. Moreover, otherwise the meaning of the parameter \alpha
would not become clear.
– In the context of this model, the text mentions "the modulation
frequency of the light wave". How and why does a modulation arise in
experiments with standing laser waves? If this is an additional
feature incorporated in these experiments, it should be explained.
– Section 3 is dedicated to systems with more than two degrees of
freedom, focusing on the role of the Arnold web. That is for sure a
fascinating issue, restricting or even excluding the occurrence of
chaos-assisted tunneling. However, the section ends inconclusively,
presenting evidence for the existence of an Arnold web in models of
systems with 2+1 freedoms, without even addressing in any way the
obvious question whether or not chaos-assisted tunneling has been
observed in these systems.
This review certainly deserves being published in Entropy, but suggest
that the author take the above remarks into account in a revised
version, in order to optimize the impact of this work.
[1] A.J. Lichtenberg, M.A. Lieberman, “Regular and Chaotic Motion”,
2nd ed., Applied Mathematical Sciences, vol. 38, Springer
(New York, 1992), Section 5.2.
[2] M. Latka, P. Grigolini, B.J. West, Phys. Rev. E 50 1071 (1994).
The text contains a number of typos that should be fixed by a careful editing.
Reviewer 2 Report
Comments and Suggestions for Authors
The manuscript presents a valuable review of the subject mentioned in the title: chaos assisted tunneling. It describes both classical and quantum dynamics, with mention to relevant modern experiments, employing a pedagogical style.
I recommend its publication after a few minor corrections:
- At the end of the introduction it is written: “For systems with one degree of freedom, there is no chaos. It requires two degrees of freedom to begin to see chaotic behavior in classical dynamical systems.” This statement is incomplete, it is valid only for energy conserving systems. It is made clear at the end of the following paragraph, which reads: “Systems with one spatial degree of freedom, driven by a time-periodic force, are also 2D systems and can become chaotc. “
The redaction of this sentences can be improved for clarity. It would be helpful to have a brief explanation of time dependence as an extra degree of freedom.
- All the references to the original publication of the figures are incomplete.
- It is difficult to understand what is being shown in Fig. 2, the description must be improved.
- In line 135 it reads "to have (i) great enhancement their average (energy) splitting", seems to miss a word.
- In line 187 it reads "studied chaos assisted tunneling in using microwave spectra", seems to have an extra "in".
- Ref. 46 starts with a point.
